# Development of a Smartphone-Based Fluorescent Immunochromatographic Assay Strip Reader

**DOI:** 10.3390/s20164521

**Published:** 2020-08-13

**Authors:** Qi Zheng, Huihuang Wu, Haiyan Jiang, Jiejie Yang, Yueming Gao

**Affiliations:** 1Zhicheng College, Fuzhou University, Fuzhou 350002, China; fdzczq@fzu.edu.cn; 2Key Lab of Medical Instrumentation & Pharmaceutical Technology of Fujian Province, Fuzhou 350108, China; N181127041@fzu.edu.cn (H.W.); jianghaiyan@fzu.edu.cn (H.J.); chy@fzu.edu.cn (J.Y.); 3College of Physics and Information Engineering, Fuzhou University, Fuzhou 350108, China

**Keywords:** fluorescence immunochromatographic assay, CMOS image sensor, smartphone, android, quantitative detection

## Abstract

Fluorescence immunochromatographic assay (FICA) is a rapid immunoassay technique that has the characteristics of high precision and sensitivity. Although image FICA strip readers have the advantages of high portability and easy operation, the use of high-precision complementary metal oxide semiconductor (CMOS) image sensors leads to an increase in overall cost. Considering the popularity of CMOS image sensors in smartphones and their powerful processing functions, this work developed a smartphone-based FICA strip reader. An optical module suitable for the test strips with different fluorescent markers was designed by replacing the excitation light source and the light filter. An android smartphone was used for image acquisition and image denoising. Then, the test and control lines of the test strip image were recognized by the sliding window algorithm. Finally, the characteristic value of the strip image was calculated. A linear detection range from 10 to 5000 mIU/mL (*R*^2^ = 0.95) was obtained for human chorionic gonadotrophin with the maximum relative error less than 9.41%, and a linear detection range from 5 to 4000 pg/mL (*R*^2^ = 0.99) was obtained for aflatoxin B1, with the maximum relative error less than 12.71%. Therefore, the smartphone-based FICA strip reader had high portability, versatility, and accuracy.

## 1. Introduction

Fluorescence immunochromatographic assay (FICA) is a common point-of-care testing (POCT) technique [1], wherein fluorescent particles are used to label antibodies and analyte concentration is quantitatively detected by analyzing the fluorescence intensity of a test strip [2,3]. Given its advantages of high stability, sensitivity, and low interference by natural fluorescence, FICA has been widely used in food safety [4], clinical diagnosis [5], and environmental monitoring [6].

With the rapid development of POCT in recent years, the development of FICA strip readers has been directed toward miniaturization, mobility, and intelligence [7]. The photoelectric scanning method or the image analysis method is mainly adopted in existing FICA strip readers. A photoelectric FICA strip reader uses a stepping motor to drive a photoelectric receiver to scan a test strip, and quantitative detection is performed by analyzing the variation rule of the analog electrical signal output by the photoelectric receiver. At present, most FICA strip readers are based on the photoelectric scanning method [8,9,10]. Although these readers have high accuracy and stability, the position of a test strip must be located before detection. Numerous works on the follow-up maintenance and debugging of photoelectric FICA strip readers for test strips of different sizes have been conducted. An image FICA strip reader uses an image sensor to collect strip images, and extracts the characteristic values of test strips through image segmentation and image processing [11,12]. In contrast to a photoelectric FICA strip reader, an image FICA strip reader lacks a stepper motor and analog signal processing circuit, and thus has the advantages of simple structure and small size. However, expensive high-performance complementary metal oxide semiconductor (CMOS) image sensors are often used to improve the detection precision of image FICA strip readers. Therefore, the use of high-performance CMOS image sensors increases overall costs.

Smartphones have evolved from simple communication tools into smart terminals that integrate multiple functions. They have the advantages of portability, powerful computing performance, and convenient data transmission. Above all, they are the largest application field of CMOS image sensors. There has been much research on quantitative detection based on smartphones. For instance, Jing Wu et al. [13] developed a magnetic lateral flow strip based on magnetic beads and smartphone camera, which converts the color density of the magnetic beads into gray values through image processing, to achieve quantitative detection of cocaine in urine samples. Considering that current smartphones are equipped with CMOS image sensors, and identifying user information and obtaining scale curves by scanning two-dimensional codes on test strips are easy, an image FICA strip reader based on smartphones was developed in this work. The smartphone-based FICA strip reader, which contains an optical module that is suitable for test strips with different fluorescent markers and uses an android smartphone to perform image acquisition, image processing, and characteristic value extraction, has high portability, versatility, and accuracy.

## 2. Development of the Proposed FICA Strip Reader

### 2.1. Principle of Quantitative Detection

FICA is a membrane detection technique for quantitative analysis [14,15]. The double-antibody sandwich immunochromatographic assay (ICA) is generally adopted when detecting macromolecular antigens with multiple antigenic determinants (proteins, viruses, and pathogenic bacteria). When the concentration of the analyte is higher, the more fluorescent particles are enriched on the T line, so the fluorescence intensity of the T line is stronger. However, the fluorescent intensity of the C line used to verify the validity of the test strip is unchanged, so the characteristic value of the test strip is larger. Competitive ICA is generally adopted when detecting micromolecule haptens with single antigenic epitopes (toxins or drug residues). When the concentration of the analyte is higher, the fewer fluorescent particles are enriched on the T line due to competitive inhibition, so the fluorescence intensity of the T line is weaker; the more fluorescent particles are enriched on the C line, so the fluorescence intensity of the C line is strong. Therefore, the characteristic value of the test strip is smaller.

Although the quantitative detection methods for different analytes are different, both the double-antibody sandwich ICA for the detection of macromolecular antigens and the competitive ICA for detection of micromolecule haptens can be realized in the test strip. For instance, Fan Yang et al. [16] developed a double-antibody sandwich ICA test strip for the rapid detection of H9 subtype of avian influenza viruses’ antibodies. Guanghua Li et al. [17] developed a competitive ICA test strip that can be used for the simultaneous detection of aflatoxin M1 and melamine in milk. For test strips prepared via double-antibody sandwich ICA, high antigen concentrations in test solutions are associated with bright test lines (T lines) and control lines (C lines) with unchanged brightness. For test strips prepared via competitive ICA, high antigen concentrations in test solution are associated with bright C lies and faded T lines.

Under illumination by an excitation light source, the fluorescence intensity of the T and C lines of a test strip is linearly related to the concentration of fluorescent particles on the T and C lines [12]. During testing, the ratio of the gray value of the T line to C line is equal to the ratio of the fluorescence intensity of the T line to the C line [18]. The T and C lines of the same test strip are labeled with the same fluorescent marker. Thus, the characteristic value (*t*) of the test strip is given by:
(1)$$t=\frac{c_{T}}{c_{C}}=\frac{I_{\mathrm{fT}}}{I_{\mathrm{fC}}}=\frac{H_{T}}{H_{C}}$$
where *c*_T_ and *c*_C_ represent the concentration of the fluorescent particles on the T and C lines, respectively; *I*_fT_ and *I*_fC_ represent the fluorescence intensity of the T and C lines, respectively; and *H*_T_ and *H*_C_ represent the gray value of the T and C lines, respectively. Therefore, for the image analysis method, the characteristic value of the test strip can be obtained by calculating the ratio of the gray value of the T line to the C line.

### 2.2. Development of the Optical Module

A single excitation light source was adopted in the design scheme, to ensure the uniformity of the light intensity of the excitation light source, and facilitate subsequent image processing. The light path should be increased through the method of light reflection to expand the illumination range of a single light-emitting diode (LED). In this work, the reflective surface of aluminum foil tape was used as the reflective material, and the back gum of the tape was easily pasted on an optical module. Aluminum foil has an ultraviolet (UV) light reflectivity of approximately 80%, and a small amount of diffuse reflection. After the excitation light was emitted by the LED, it was reflected by the aluminum foil to the observation window of the test strip. It finally reached the camera of the mobile phone through a light filter and a macro lens.

The interference of external light must be eliminated, and the whole optical system should be as closed as possible, to ensure the consistency of the test environment. The optical module was constructed from light-cured acrylonitrile butadiene styrene plastic by using a 3D printer, and opaque materials were pasted around and in the bottom plate to isolate external light. The thickness of the outer shell and the partition of the optical module was 2 mm, and the sites where the accessories were installed were thickened and strengthened as needed.

The optical module was powered by universal serial bus, and provided a stable excitation current to the LED through a constant-current source module. Its excitation light source and light filter could be selected in accordance with the excitation and emission spectra of the fluorescent particles for compatibility with the test strips with different fluorescent markers. In general, the minimum clear imaging distance of smartphones is between 6.5 and 7.5 cm. A 15-fold macro lens, which could reduce the minimum clear imaging distance of the smartphones to approximately 3 cm, and enable the clear shooting of objects at close range, was installed at the viewfinder, to reduce the size of the optical module. Under the condition that the mobile phone could focus accurately, the size of the optical module should be minimized to facilitate transport. The overall size of the optical module was 115 mm long, 70 mm wide, and 45 mm high. The design process of the optical module is shown in Figure 1.

### 2.3. Image Processing and Characteristic Value Extraction

Compared with computers, smartphones have poorer computing power and more roles during use. Therefore, the software algorithm should be designed to minimize the occupation of system resources, to improve software running speed. The sliding window algorithm, instead of the clustering algorithm used by computers to process strip images, was used in the recognition of the T and C lines of the strip image, as shown in Figure 2.

If a valid sample image is obtained when the program is run, the image of the observation window of the test strip is first intercepted, and then image processing is performed. This approach not only greatly reduces the calculation amount of the smartphone, but also increases program speed. Then, the image of the observation window of the test strip is separated into two areas with the same sizes on the left and right, and two rectangular windows with the same size (*i* × *j*) are used as the sliding window. The effect of the sliding window size on the calculation of the characteristic value is shown in the detection of the standard strip images in Section 3.1. Two sequences, namely, {*H*_x_}, {*H*_y_}, can be obtained by calculating the gray value of each position in the sliding process of each sliding window.

The gray value of the sliding window on the left side is:
(2)$$H_{x}=\sum_{x}^{x+i}\sum_{0}^{j}h(x,j)$$


The gray value of the sliding window on the right side is:
(3)$$H_{y}=\sum_{y}^{y+i}\sum_{0}^{j}h(y,j)$$
where *x* is the horizontal coordinate of the left side, *y* is the horizontal coordinate of the right side, and *h*(*x*, *j*) is the gray value of point (*x*, *j*). The brightest parts on the left and right sides correspond to the T and C lines of the test strip, respectively. Thus, the gray values of the T and C lines are given by:
(4)$$H_{c}=\max\{H_{x}\}$$
(5)$$H_{T}=\max\{H_{y}\}$$


Substituting (4) and (5) into (1) yields
(6)$$t=\frac{H_{T}}{H_{C}}=\frac{\max\{H_{y}\}}{\max\{H_{x}\}}$$


Although a light filter was installed in front of the macro lens, some fluorescent particles remained in the background area of the test strip, due to the insufficient chromatographic assay during the reaction. Therefore, the interference of the background gray value should be subtracted when calculating the gray values of the T and C lines. If the gray value of the background area is HB, then the calculation formula of the characteristic value is given by:
(7)$$t=\frac{H_{T}-H_{B}}{H_{C}-H_{B}}=\frac{\max\{H_{y}\}-H_{B}}{\max\{H_{x}\}-H_{B}}$$


When using double-antibody sandwich ICA for the quantitative detection of an analyte, such as human chorionic gonadotrophin (HCG), high analyte concentration is associated with the high gray value of the T line, and the gray value of the C line remains unchanged. Therefore, the characteristic value of the test strip is positively correlated with the concentration of the analyte. When using competitive ICA for the quantitative detection of an analyte, such as aflatoxin B1 (AFB1), high analyte concentration is associated with the small gray value of the T line and the large gray value of the C line. Therefore, the characteristic value of the test strip is negatively correlated with the concentration of the analyte.

For adaption to test strips of different sizes, a calibration program for the screenshot frame was set in the software. This program could calibrate the size and position of the screenshot frame, in accordance with the actual position of observation window of the test strip in the optical module. The actual test showed that the T and C lines of the test strip were evenly distributed in the vertical direction, and the site outside the observation window was the background. Therefore, when the size of the screenshot frame was slightly smaller than the observation window of the test strip, the manual difference of the calibration operation had a negligible effect on the accuracy of the quantitative detection. The steps of image processing were as follows:
(1)When the original image is collected, the android smartphone reads the calibration parameters of the screenshot frame and writes them into the screenshot function. Then, the strip image is cut to obtain the image of the observation window of the test strip.(2)The filter function of the OpenCV computer vision library is used to reduce the noise, which is caused by the interference of environment light, sensor temperature, light source, and other factors during collection and transmission, of the strip image.(3)The processed image is divided into two equal left-hand and right-hand areas, and two rectangular sliding windows of the same size are established. The two sliding windows traverse their respective regions, with one pixel as the step length. The maximum gray value of the window coverage in their respective areas is calculated.(4)The gray value when the sliding window is horizontally centered is taken as the background gray value. The maximum gray value obtained for the sliding window in the left-hand and right-hand areas is subtracted from the background gray value, and then the ratio of the two difference values is taken as the characteristic value of the strip image.


## 3. Experiment

### 3.1. Detection of Standard Strip Images

The standard strip images shown in Figure 3a were used for performance verification in this work, to verify the effect of the proposed FICA strip reader on image processing and the characteristic value extraction of the test strip. The smartphone-based FICA strip reader could accurately recognize the positions of the T and C lines of the test strip in the detection of the standard strip images.

As found in the actual test, the T and C lines were approximately 60 pixels in width. Therefore, the characteristic values of each strip with the width of the sliding window within the range of 20–60 pixels were calculated at 20-pixel intervals. The test results are shown in Figure 3b. The calculated characteristic values of each test strip under different sliding window widths were very close. Therefore, when the width of the sliding window was smaller than the widths of the T and C lines, the change in the width of the sliding window had little effect on the detection results.

### 3.2. Quantitative Detection of HCG 

Rare elements were used as the fluorescent marker in the HCG test strip (Triplex International Biosciences Co., LTD., Xiamen, China) in this experiment, and their excitation and emission spectra are shown in Figure 4. The UV LED model UVTOP335TO39HS with the peak light emission wavelength of 340 nm was selected as the excitation light source of the HCG test strip, and a light filter with 610 nm wavelength was installed in front of the macro lens.

#### 3.2.1. Detection of HCG Test Strips

A part of HCG test strips and the collected images in the experiment were shown in Figure 5. The brightness of T line, C line and background area in the observation window of fluorescent test strip was uniform.

#### 3.2.2. Curve Fitting

The proposed FICA strip reader was used to detect HCG solutions with 9 different concentrations (10, 25, 50, 125, 250, 500, 1250, 2500, and 5000 mIU/mL). HCG solutions were prepared by diluting HCG powder (TanMo Quality Testing Technology Co., LTD., Beijing, China) with deionized water. The test strip of each concentration was tested 6 times, and the average value of the test results was taken as the characteristic value. The least square method was used for curve fitting, to obtain the relationship between the concentration of the HCG solution and the characteristic value of the HCG test trip detected by the proposed FICA strip reader. This relationship is:
(8)t=0.132ln(c)+0.120
where *c* is the concentration of the HCG solution, and *t* is the characteristic value of the HCG test strip detected by the proposed FICA strip reader. As can be seen from Figure 6, a linear detection range from 10 to 5000 mIU/mL (*R*^2^ = 0.95) was obtained. The limit of detection (LoD) was 4.7 mIU/mL with 3δ/slope calculation. Compared with the optical inspection system for HCG detection based on the Taguchi method proposed in reference [19], which has a linear range of 6.25–50.00 mIU/mL (*R*^2^ = 0.99) and a detection limit of 6.25 mIU/mL, our FICA strip reader has a wider application range, due to its wider linear detection range.

#### 3.2.3. Accuracy Test

The proposed FICA strip reader was used to detect HCG solutions with 5 different concentrations (25, 50, 500, 2500, and 5000 mIU/mL). By substituting the characteristic value t of the test strip image detected by the proposed FICA strip reader into (8), HCG concentration, recovery rate, and relative error could be calculated. The experimental data are shown in Table 1.

The test results in the above table showed that the relative error of the test result for the 5000 mIU/mL HCG solution was the largest (9.41%). This result indicated that the smartphone-based FICA strip reader had high accuracy in the quantitative detection of HCG solutions.

### 3.3. Quantitative Detection of AFB1 

In this experiment, phycocyanin was used as the fluorescent marker in the AFB1 test strip (laboratory self-made), and its excitation and emission spectra are shown in Figure 7. The LED model UVTOP335TO39HS with the peak light emission wavelength of 555 nm was selected as the excitation light source of the AFB1 test strip, and a light filter with a wavelength of 640 nm was installed in front of the macro lens.

#### 3.3.1. Detection of AFB1 Test Strips

A part of AFB1 test strips and the collected images in the experiment were shown in Figure 8. The brightness of T line, C line and background area in the observation window of fluorescent test strip was uniform.

#### 3.3.2. Curve Fitting

The proposed FICA strip reader was used to detect AFB1 solutions with 9 different concentrations (5, 50, 100, 250, 600, 800, 1000, 2000, and 4000 pg/mL). AFB1 solutions were prepared by diluting AFB1 powder (Shanghai Yuduo Biotechnology Co., LTD., Shanghai, China) with deionized water. The test strip of each concentration was tested 6 times, and the average value of the test results was taken as the characteristic value. The least square method was used for curve fitting, to obtain the relationship between the concentration of the AFB1 solution and the characteristic value of the AFB1 test trip detected by the proposed FICA strip reader. This relationship is:
(9)t=−0.401ln(c)+3.328
where *c* is the concentration of the AFB1 solution, and *t* is the characteristic value of the AFB1 test strip detected by the proposed FICA strip reader. As can be seen from Figure 9, a linear detection range from 5 to 4000 pg/mL (*R*^2^ = 0.99) was obtained. The LoD was 3.9 pg/mL with 3δ/slope calculation. Compared with the smartphone-based fluorimetric sensor system for AFB1 detection proposed in reference [20], which has a linear range of 20–100 ng/mL (*R*^2^ = 0.98), and a detection limit of 20 ng/mL, our FICA strip reader has a wider application range due to its lower LoD.

#### 3.3.3. Accuracy Test

The proposed FICA strip reader was used to detect AFB1 solutions with 5 different concentrations (25, 75, 500, 1000, and 2000 pg/mL). AFB1 concentration, recovery rate, and relative error could be calculated by substituting the characteristic value t of the test strip image detected by the proposed FICA strip reader into (9). The experimental data are shown in Table 2.

The test results in the above table showed that the relative error of the test result for 1000 pg/mL AFB1 solution was the largest (12.71%). This result showed that the smartphone-based FICA strip reader had high accuracy in the quantitative detection of AFB1 solutions.

## 4. Conclusions

With the popularization of smartphones, detection devices based on smartphones have shown the advantages of portability, easy operation, and low cost. These characteristics are in line with the development trend of POCT. By analyzing the shortcomings of current FICA strip readers, this work developed a smartphone-based FICA strip reader.

The proposed FICA strip reader consisted of an optical module and an android smartphone. The optical module used the reflective light of a single LED to improve the uniformity of the light source and facilitated subsequent image processing. The excitation light source and the filter could be replaced in accordance with the characteristics of the fluorescent particles. The use of a macro lens shortened the minimum clear imaging distance, and reduced the size of the optical module. The smartphone was used for image collection and image processing and for the characteristic value extraction of the test strip. The sliding window algorithm was used to recognize the T and C lines of the test strip. This approach reduced the occupation of system resources and increased the speed of the software. In the quantitative detection of HCG, the linear detection range was 10 to 5000 mIU/mL (*R*^2^ = 0.95), with the maximum relative error less than 9.41%, and the LoD of the test strip for HCG was 4.7 mIU/mL. In the quantitative detection of AFB1, the linear detection range was 5 to 4000 pg/mL (*R*^2^ = 0.99), with the maximum relative error less than 12.71%, and the LoD of the test strip for AFB1 being 3.9 pg/mL. The smartphone-based FICA strip reader has broad application prospects in POCT, given its high portability, versatility, and accuracy.

## Figures and Tables

**Figure 1 sensors-20-04521-f001:**
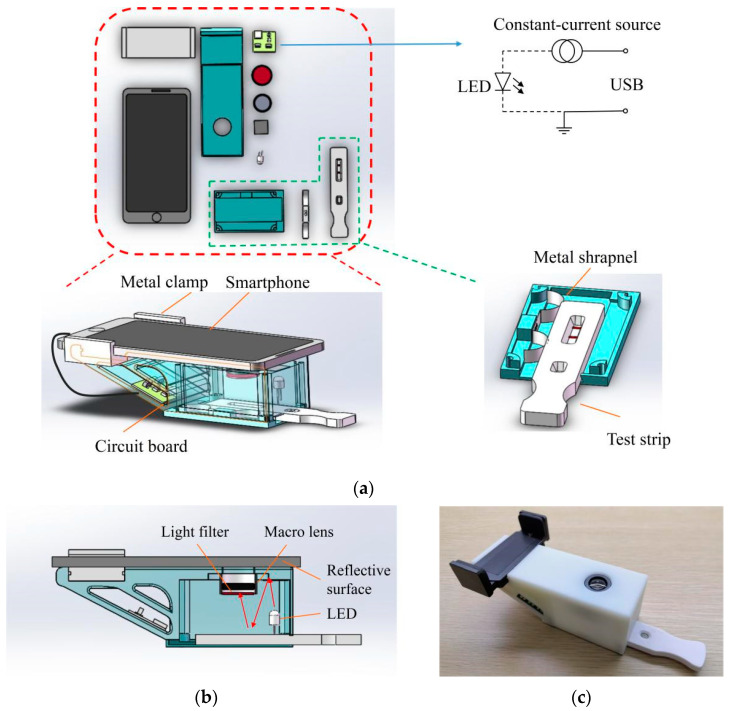
(**a**) Design diagram of the optical module; (**b**) Optical path of the optical module; (**c**) Physical diagram of the optical module.

**Figure 2 sensors-20-04521-f002:**
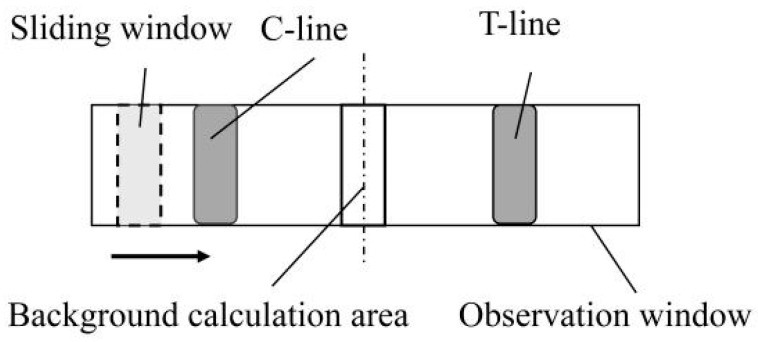
Schematic of sliding window detection.

**Figure 3 sensors-20-04521-f003:**
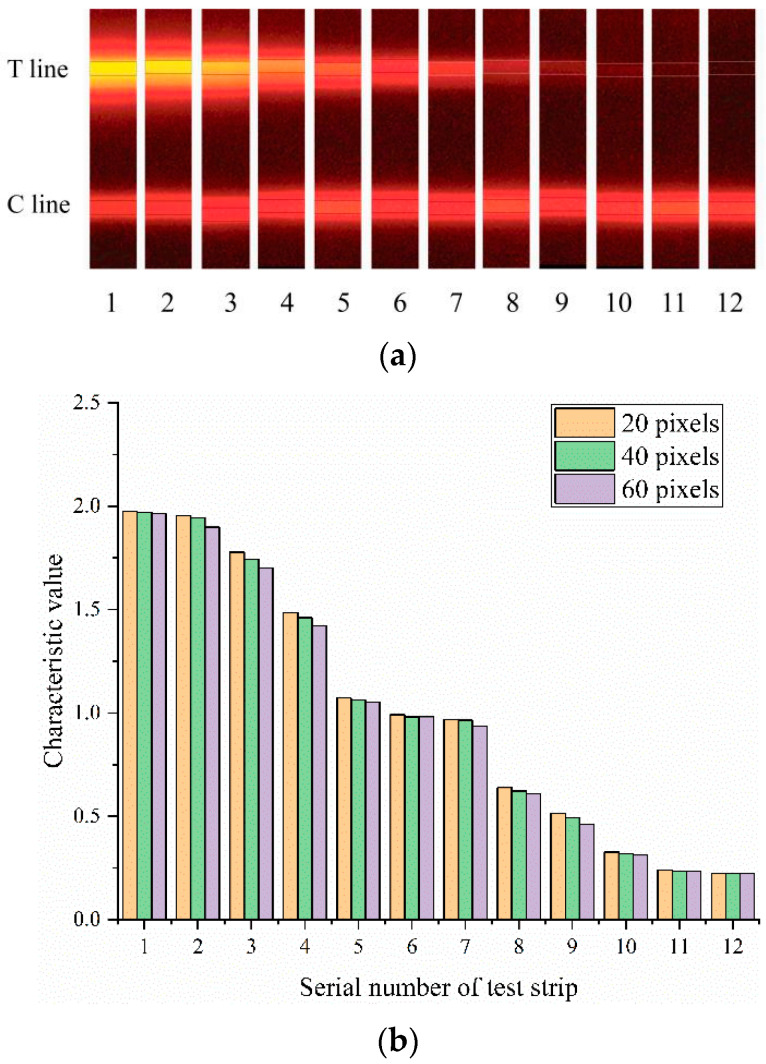
(**a**) Standard strip images; (**b**) Test results of the standard strip images.

**Figure 4 sensors-20-04521-f004:**
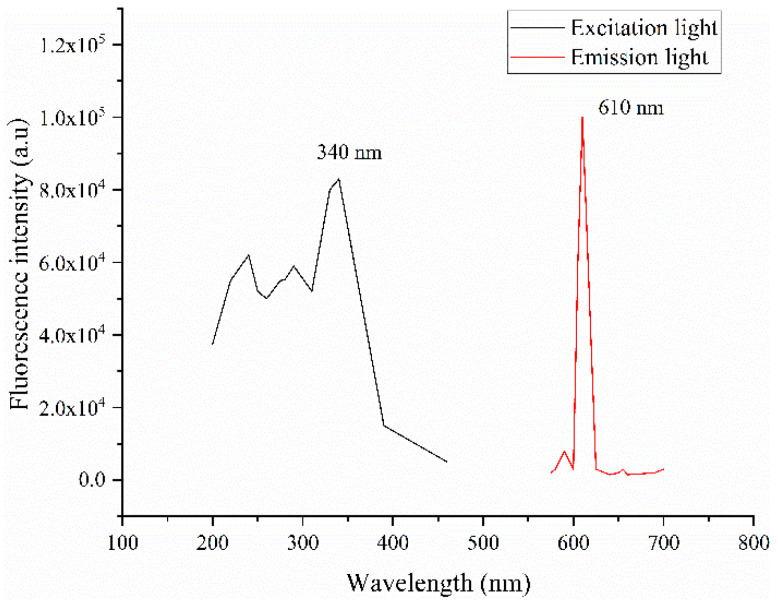
Excitation and emission spectra of the human chorionic gonadotrophin (HCG) test strip.

**Figure 5 sensors-20-04521-f005:**
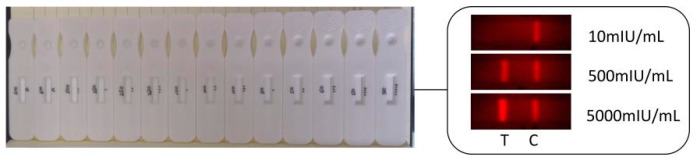
HCG test strips and collected images.

**Figure 6 sensors-20-04521-f006:**
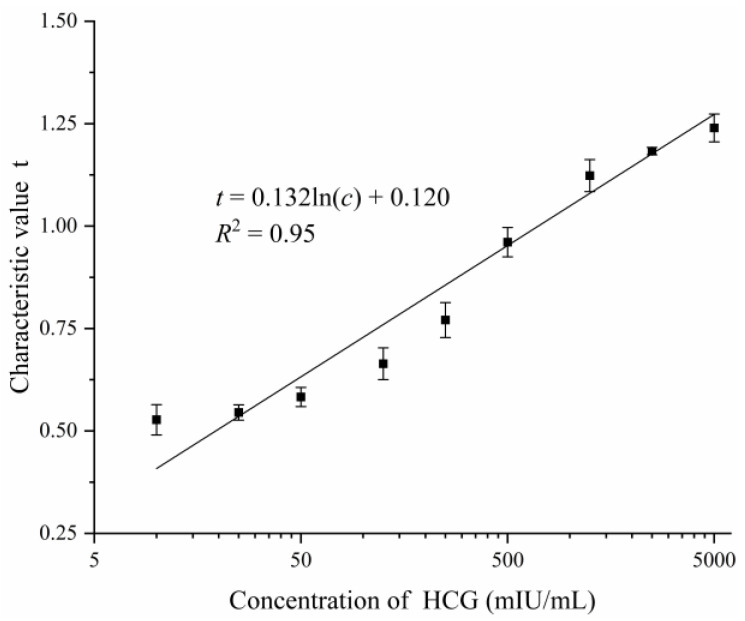
Relationship between the concentration of HCG and characteristic value.

**Figure 7 sensors-20-04521-f007:**
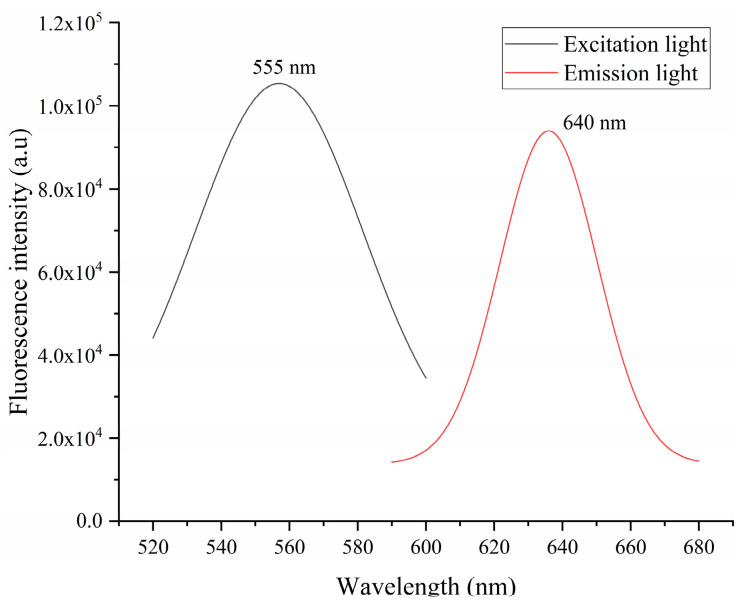
Excitation and emission spectra of the aflatoxin B1 (AFB1) test strip.

**Figure 8 sensors-20-04521-f008:**
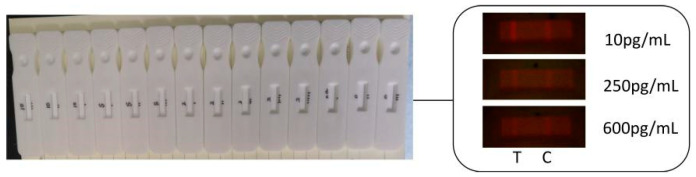
AFB1 test strips and collected images.

**Figure 9 sensors-20-04521-f009:**
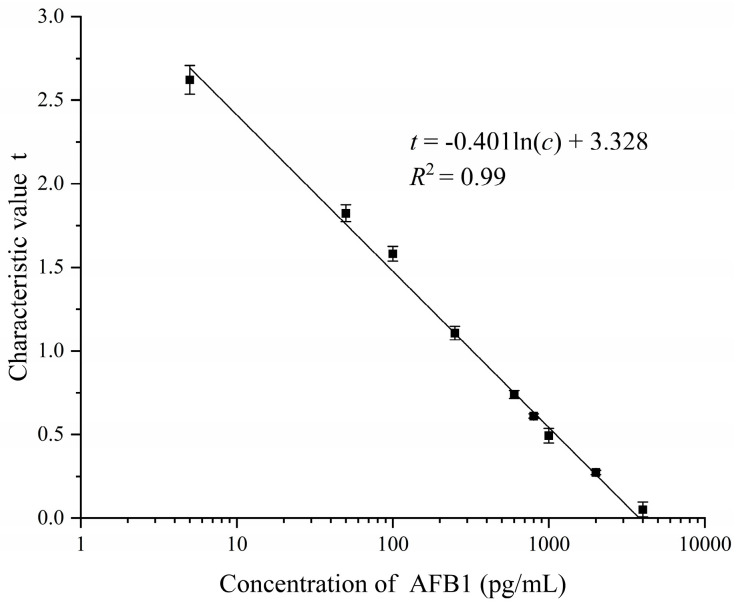
Relationship between the concentration of AFB1 and characteristic value.

**Table 1 sensors-20-04521-t001:** Test results for HCG.

Concentration (mIU/mL)	Characteristic Value	Calculated Concentration (mIU/mL)	Recovery Rate	Relative Error
25	0.547	25.532	102.13%	2.13%
50	0.623	45.468	90.94%	9.06%
500	0.935	485.913	97.18%	2.82%
2500	1.161	2702.818	108.11%	8.11%
5000	1.229	4529.537	90.59%	9.41%

**Table 2 sensors-20-04521-t002:** Test results for AFB1.

Concentration (pg/mL)	Characteristic Value	Calculated Concentration (pg/mL)	Recovery Rate	Relative Error
25	2.069	23.096	92.38%	7.62%
75	1.585	77.218	102.96%	2.96%
250	0.999	278.572	111.43%	11.43%
1000	0.510	1127.132	112.71%	12.71%
2000	0.270	2050.697	102.53%	2.53%

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
