# Peer review of "Development of a Smartphone-Based Fluorescent Immunochromatographic Assay Strip Reader"

_sensors, 2020, doi:10.3390/s20164521_

Round 1

Reviewer 1 Report

The manuscript developed a fluorescence immunochromatographic assay with smartphones. It is an interesting work from the view of smartphone reader. Authors described how to design a suitable optical module with different excitation light source and the light filter, and how to acquire image and reduce noise via Android. However, there are several key issues should be addressed.

Major

  1. The references does not cover the major work in this area. The following references should be included. (1) On-Site Ultrasensitive Detection Paper for Multiclass Chemical Contaminants via Universal Bridge-Antibody Labeling: Mycotoxin and Illegal Additives in Milk as an Example. ANALYTICAL CHEMISTRY, 2019, 91, 1968-1973. (2) Magnetic Lateral Flow Strip for the Detection of Cocaine in Urine by Naked Eyes and Smart Phone Camera. SENSORS, 2017, 17, 1286.
  2. 1. Principle of quantitative detection. It should be make clear that either sandwich immunoassay for protein, antigen, etc or competitive immunoassay for the small molecules, such as aflatoxin can be realized in the test strip. Authors should include the principle of both immunoassay in the section. In the sandwich immunoassay, the signal raise according to the concentration of target of interest, while in the competitive immunoassay, the signal reduced as the concentration of small molecule goes up.
  3. The limit of detection should be calculated.
  4. The result from this smartphone based detection should be compared with other published papers.
  5. Fig 5 and 7, add the error bar from at least triple experiments.

Author Response

Reviewer#1, Concern # 1: The references does not cover the major work in this area.

Author response: Thanks for your comments. We have added appropriate references. (see P2 lines  55-59 and 82-85)

Reviewer#1, Concern # 2: It should be make clear that either sandwich immunoassay for protein, antigen, etc or competitive immunoassay for the small molecules, such as aflatoxin can be realized in the test strip.

Author response: Thanks for your comments. We have made the supplement. (see P2 lines 80-82)

Reviewer#1, Concern # 3: Authors should include the principle of both immunoassay in the section.

Author response: Thanks for your comments. We have further supplemented the principles of the two immunoassays. (see P2 lines 71-74 and lines 75-79)

Reviewer#1, Concern # 4: The limit of detection should be calculated.

Author response: Thanks for your comments and we have calculated the limit of detection in the experimental section. (see P8 lines 226-227 and P10 lines 264-265)

Reviewer#1, Concern # 5: The result from this smartphone based detection should be compared with other published papers.

Author response: Thanks for your comments. We have compared the test results of HCG and AFB1 with other published papers. (see P8 lines 227-230 and P10 lines 265-267)

Reviewer#1, Concern # 6: Fig 5 and 7, add the error bar from at least triple experiments.

Author response: Thanks for your comments. We have added error bars to the original Fig. 5 and Fig. 7. They are now Fig. 6 and Fig. 9, respectively. (see the top of P8 and P10)

Reviewer 2 Report

General Comments:

    This paper described that development of a smartphone-based fluorescent immunochromatographic assay strip. There are some points need to be clarified.

1. Why do use the different fluorescent reagents in the HCG and AFB1 strip system?

2.Is the Figres 3A a real pictures?

3.This developed smartphone-based fluorescent immunochromatographic assay strip manuscript lacks of real sample testing.

4. Please show the pictures of AFB1 fluorescent immunochromatographic assay strip.

Specific comments:

P2 line 71 ... ..dark T lines-->..fade T line.

P5 line 140..  the gray value of the T line remains unchanged-->..the gray value of the C line

P9 line 236 …1000 mIU/mL--> 1000 pg/mL

Table1  The third column Calculated concentration (pg/mL)-->Concentration                (mIU/mL)

Table 2.  The First column Concentration (mIU/mL)-->Concentration (pg/mL)

Author Response

Reviewer#2, Concern # 1: Why do use the different fluorescent reagents in the HCG and AFB1 strip system?

Author response: Thanks for your comments. The proposed optical module has the ability to detect the fluorescence markers with different wavelength. Therefore, we verified this through experiments. Due to the use of different fluorescent markers, the wavelength of the excitation light source required for the experiment is also different. Therefore, we realized the detection of test strips with different fluorescent markers by switching the wavelength of the excitation light source.

Reviewer#2, Concern # 2: Is the Figres 3A a real pictures?

Author response: Thanks for your comments. Fig. 3(a) is a standard picture that we made according to the intensity of the fluorescent emission light of the test strip and specially used for the calibration of the optical module and the smartphone App.

Reviewer#2, Concern # 3: This developed smartphone-based fluorescent immunochromatographic assay strip manuscript lacks of real sample testing.

Author response: Thanks for your comments. At present, we are mainly engaged in the research of fluorescence detection devices based on smartphones. We use standard samples for testing in our experiments, which verifies the testing capabilities of our strip reader. At the same time, given the impact of the COVID-2019 in the world, it is very difficult for us to collect relevant samples at this stage.

Reviewer#2, Concern # 4: Please show the pictures of AFB1 fluorescent immunochromatographic assay strip.

Author response: Thanks for your comments. We have added pictures in the experiment section. (see P7 Fig. 5 and P9 Fig.8)

Reviewer#2, Concern # 5: Specific comments:

P2 line 71 ... ..dark T lines-->..fade T line.

P5 line 140..  the gray value of the T line remains unchanged-->..the gray value of the C line

P9 line 236 …1000 mIU/mL--> 1000 pg/mL

Table1  The third column Calculated concentration (pg/mL)-->Concentration                (mIU/mL)

Table 2.  The First column Concentration (mIU/mL)-->Concentration (pg/mL)

Author response: Thanks very much for your carefulness to put forward the mistakes in the paper. We have modified them in the manucript. They are now  in the following positions: P3 line 89, P5 line 160, P10 line 275, P8 Table1 and P10 Table 2.

Reviewer 3 Report

Dear author(s),

You describe an interesting new device for analytic purposes. The technique is very complex and difficult to understand for readers that are not familiar.

Further, I have some remarks:

line 22: the term goodness of fit is correct but is not commonly used.

line 48: I think you forgot to mention that HP-CMOS is an expensive method.

line 63: FICA is not a very new technique.

line 147: do you mean the strips may differ in size depending on the manufacturere?

line 153: what means 'manual difference'?

Finally, they details of the samples used are missing. For example, composition, source of the materials, etc.

In addition, there are no comparisons with existing assays.

Maybe after some corrections/adaptations this manuscript can be published.

Author Response

Reviewer#3, Concern # 1: Line 22: the term goodness of fit is correct but is not commonly used.

Author response: Thanks for your comments. We agree with your opinion and made changes in the manucript. (see P1 lines 22-25, P8 lines 225-226, P10 lines 263-264,P11 lines 290-294)

Reviewer#3, Concern # 2: Line 48: I think you forgot to mention that HP-CMOS is an expensive method.

Author response: Thanks for your kind reminding. We have made the supplement. (see P2 lines 49-51)

Reviewer#3, Concern # 3: Line 63: FICA is not a very new technique.

Author response: Thanks for your comments. We agree with your opinion and made changes in the manucript. (see P2 line 68)

Reviewer#3, Concern # 4: Line 147: do you mean the strips may differ in size depending on the manufacture?

Author response: Thanks for your comments. The difference in the size of the test strips we are referring to is caused by the different sizes of the shells of the test strips manufactured by different manufacturers. Therefore, you can see in Fig. 1(a) that we have added a metal shrapnel in the optical module to adapt to the possible subtle differences in the size of the test strips of different manufacturers.

Reviewer#3, Concern # 5: Line 153: what means 'manual difference'?

Author response: Thanks for your comments. Since the test strip size may be various in practise, the relative position of the observation window of the test strip in the optical module is not fixed. Therefore, we designed a calibration program for the screenshot frame. We can open the mobile camera to determine the position of the observation window of the test strip, and then manually select the size and position of the screenshot frame, so that our FICA reader can be applied to different sizes of test strips.

Reviewer#3, Concern # 6: The details of the samples used are missing.

Author response: Thanks for your comments. We have added them in the manuscript. They are respectively in the following positions: P7 lines 203-204, P7 lines 216-218, P8 line 242 and P9 lines 254-256.

Reviewer#3, Concern # 7: There are no comparisons with existing assays.

Author response: Thanks for your comments. We have compared the test results of HCG and AFB1 with other existing assays. (see P8 lines 227-230 and P10 lines 265-267)

Round 2

Reviewer 1 Report

This work was revised well and could be published as is.

Author Response

Thank you very much for your comments.

Reviewer 2 Report

 At the same time, given the impact of the COVID-2019 in the world, it is very difficult for us to collect relevant samples at this stage--> There is no correlation.

Author Response

Reviewer#2, Concern # 1: At the same time, given the impact of the COVID-2019 in the world, it is very difficult for us to collect relevant samples at this stage--> There is no correlation.

Author response: Thanks for your comments. In this work, we focused on developing a universal FICA strip reader based on the smartphone. A series of standard solutions were prepared to verify the performance of the proposed device. Since the concentration of the standard solution is known, it can directly reflect the accuracy and the repeatability of the detection results. Then the limit of detection and linear detection range of the test strip can be determined.

The HCG test strip used in this experiment is a commercial kit certificate by the SFDA. Adequate clinical trials have been carried out prior to certification. It can be seen that the test strip has good detection performance. Its repeatability CV ≤ 15.0%, relative deviation of accuracy ≤ ±15.0%, and recovery within 85.0% - 115.0%. 

For the FICA test, the strip looks like the biochemical sensor in the front end. The strip has the ability not only to detect the concentration of the analyte, but also to filter out various biochemical interference during the reaction. According to the good performance of the strip itself, the interference factors in real samples has be effectively filtered out, and will not be reflected in the fluorescence image. Therefore, choosing a standard solution with a certain gradient is sufficient to verify the detection capability of the proposed device.

Reviewer 3 Report

Dear author(s),

The manuscript has been greatly improved. In the current form it can be published.

Author Response

Thank you very much for your comments.